# Causes and Mechanisms of Hematopoietic Stem Cell Aging

**DOI:** 10.3390/ijms20061272

**Published:** 2019-03-13

**Authors:** Jungwoon Lee, Suk Ran Yoon, Inpyo Choi, Haiyoung Jung

**Affiliations:** 1Environmental Disease Research Center, Korea Research Institute of Bioscience and Biotechnology (KRIBB), 125 Gwahak-ro, Yuseong-gu, Daejeon 34141, Korea; jwlee821@kribb.re.kr; 2Immunotherapy Research Center, Korea Research Institute of Bioscience and Biotechnology (KRIBB), 125 Gwahak-ro, Yuseong-gu, Daejeon 34141, Korea; sryoon@kribb.re.kr; 3Department of Functional Genomics, University of Science and Technology (UST), 113 Gwahak-ro, Yuseong-gu, Daejeon 34113, Korea

**Keywords:** hematopoietic stem cell aging, rejuvenation, self-renewal, differentiation

## Abstract

Many elderly people suffer from hematological diseases known to be highly age-dependent. Hematopoietic stem cells (HSCs) maintain the immune system by producing all blood cells throughout the lifetime of an organism. Recent reports have suggested that HSCs are susceptible to age-related stress and gradually lose their self-renewal and regeneration capacity with aging. HSC aging is driven by cell-intrinsic and -extrinsic factors that result in the disruption of the immune system. Thus, the study of HSC aging is important to our understanding of age-related immune diseases and can also provide potential strategies to improve quality of life in the elderly. In this review, we delineate our understanding of the phenotypes, causes, and molecular mechanisms involved in HSC aging.

## 1. Introduction

In the hematopoietic system, hematopoietic stem cells (HSCs) continuously replenish the blood cells including B and T lymphocytes, erythrocytes, myeloid cells, platelets, natural killer (NK) cells, mast cells, and dendritic cells (DCs), throughout the lifetime of an organism [1,2,3]. HSCs were the first stem cells to be identified and isolated and remain the most-studied tissue-specific stem cells. HSCs constitute the pool of long-term HSCs (LT-HSCs), short-term HSCs (ST-HSCs), and multipotent progenitors (MPPs). They can be identified with specific cell-surface markers using fluorescence-activated cell sorting (FACS) technology. All murine HSCs are lineage (Lin^−^), stem cell antigen-1 (Sca-1)^+^, and cKit^+^ (LSKs) that can be characterized as more or less primitive with CD150 (Slamf1), CD48 (Slamf2), CD34, and Flt3 [4,5]. Human HSCs can also be isolated and identified by the expression of cell surface markers such as Lin^−^, CD34^+^, CD38^−^, Thy1.1^+^, and CD45RA^−^ [6,7,8] (Figure 1).

HSCs have the ability to self-renew and differentiate into immune cells; however, similar to other adult stem cells, HSCs are vulnerable to age-related stress [7,8]. With aging, HSCs gradually lose their self-renewal capacity and reconstitution potential and are therefore different from pluripotent embryonic stem cells (ESCs) and induced pluripotent stem cells (iPSCs) [9]. HSC aging is driven by both cell-intrinsic and -extrinsic factors. Although the functional change of HSCs with aging is mostly regulated by cell-intrinsic factors, including DNA damage, reactive oxygen species (ROS), epigenetic changes and polarity changes, it is also known that the hematopoietic niche or microenvironment-derived cell-extrinsic factors are essential for the maintenance of HSCs [7,10]. Hematopoietic aging alters the immune system, inducing many types of immune diseases including both myeloid and lymphoid leukemias, anemia, declining adaptive immunity, autoimmunity, increased susceptibility to infectious diseases, and vaccine failure [11,12]. Thus, the study of HSC aging is important to our understanding of age-related immune diseases and can also provide potential strategies to improve quality of life in the elderly. In this review, we summarized the hallmarks, causes and mechanisms, and rejuvenation of HSC aging, and also introduced recent emerging technologies for HSC study.

## 2. Hallmarks of HSC Aging

### 2.1. Defect in Repopulating Capacity

Though we know that the number of HSCs in bone marrow (BM) increases 2- to 10-fold with aging in both mice and humans [12,13,14], the mechanisms by which such increases in HSC numbers occur in aged organisms remain elusive. One hypothesis proposes that the increase in the number of aged HSCs is a compensatory mechanism to overcome their loss of function in normal hematopoiesis [12]. Another study suggested that an increase in the frequency of self-renewing symmetric cell divisions might contribute to the increased numbers and impaired function of aged HSCs [15]. Recent reports show that aged HSCs are less quiescent, with larger fractions undergoing cell division than young HSCs [8,16], having accumulated more oxidative DNA damage [17]. These reports go on to suggest that the function of aged HSCs may be limited by these factors. To determine the functional differences between young and aged HSCs, a competitive transplantation assay was developed as a gold standard test to assess the long-term self-renewal and multi-lineage potential of HSCs. In this method, HSCs are mixed with young BM cells that are able to restore immunity in the recipient animal post-irradiation [3,5,11,12]. We and other research groups have reported that aged HSCs exhibited a functional decline in repopulation capacity [13,18,19]. These results imply that the increased number of aged HSCs does not compensate for their loss of function, thereby resulting in immune cell homeostasis defects in aged organisms [12,15].

### 2.2. Defect in Homing and Increase in Mobilization

HSCs reside in the marrow cavity of long bones during adult life, co-existing in close association with many other cell types in BM in a highly organized structure supported by stromal cells referred to as the niche [20,21,22]. The engraftment of HSCs into nonmyeloablative hosts resulted in spatial localization of stem cell “niches,” while other transplanted BM cells were detected as flattened bone lining cells on the periosteal bone surface [23]. Osteoblastic cells constitute an important cellular component of the HSC niche [24,25]. Other groups assessed that different hematopoietic cell subsets are localized to distinct areas according to their stage of differentiation using live-imaging techniques [20,26,27]. Interestingly, transplanted HSCs tended to home to the endosteum in irradiated recipients but were randomly distributed and unstable in nonirradiated mice [27]. The homing ability of HSCs, involving the trafficking of a transplanted donor HSC to the BM of a recipient, is critical in BM transplantation procedures for the successful treatment of many blood disorders and malignancies including leukemia, lymphoma, and myeloma [28]. Liang and colleagues found that the BM homing efficiency of aged mouse HSCs was approximately three-fold lower than that of young HSCs; we also reported a reduction in the homing ability of aged HSCs in BM [1,19,29].

Systemic administration of cytokines and chemokines or cytotoxic agents mobilizes hematopoietic stem and progenitor cells (HSPCs) from the BM into peripheral blood, where they are collected in clinically useful quantities for stem cell therapies [22]. Granulocyte colony-stimulating factor (G-CSF) mobilizes hematopoietic cells from the marrow into circulation, with increased progenitor cells of all lineages detected in the spleens of G-CSF-treated mice [30,31]. In a G-CSF-receptor (G-CSFR)-deficient mice study, they found that expression of the G-CSFR on BM cells is required for mobilization of hematopoietic progenitor cells (HPCs), but this is not dependent on the expression of the G-CSFR on hematopoietic progenitor cells (HPCs). From these results, they suggested that the indirect effect of G-CSF on hematopoietic cells is essential for HSC mobilization [30,32]. G-CSF causes transient upregulation of stromal cell-derived factor-1 (SDF-1) and subsequently activates CXC chemokine receptor-4 (CXCR4) signaling for the production of hepatocyte growth factor (HGF). HGF binds to c-Met and thus activates c-Met signaling to regulate the mTOR/FOXO3a signaling pathway. Finally, G-CSF signaling causes ROS production and promotes the egress of HSCs out of the BM [31,33]. Xing and colleagues revealed that the mobilization of HSPCs from BM to peripheral blood in response to G-CSF requires the de-adhesion of HSPCs from the niche. This ability to mobilize HSCs is approximately five-fold greater in aged mice [22].

### 2.3. Lineage Skewing

Under normal conditions, HSCs differentiate into balanced myeloid and lymphoid lineages. However, aged adults show a higher prevalence of anemia and compromised adaptive immunity, due to reduced HSC number and function, caused by thymic involution and aged lymphoid progenitors [34,35]. Aging forces the differentiation of HSCs to myeloid bias, characterized by a higher percentage of myeloid cells in peripheral blood, both upon transplantation and at steady state [1,8,29,36]. Sudo and colleagues showed that although aged HSCs exhibit abnormal differentiation, they were able to regenerate blood cells over long periods of time and could self-renew. These aged HSCs exhibited myeloid biased differentiation [3,15]. This characteristic is also reflected in the relative expansion of myeloid progenitor numbers in aged compared to young mice—a feature of aged HSCs known to be a cell autonomous function [37]. The lineage skewing associated with HSC aging has been linked to an upregulation of myeloid-specific genes and a downregulation of lymphoid-specific genes [38,39].

## 3. Causes and Mechanisms of HSC Aging

### 3.1. DNA Damage

The DNA damage response (DDR) in cells is involved in cell cycle regulation, cell death and senescence, transcriptional regulation, as well as chromatin remodeling [40]. HSCs are also exposed to DDR stress due to their continuous self-renewal process as well as their microenvironment during aging. It has been proposed that DNA damage may be a principal mechanism regulating age-dependent stem cell decline; furthermore, the accumulation of genomic instability has been implicated in hematopoietic malignancy, possibly derived from transformed HSCs [40,41]. Researchers have reported that HSCs experience genomic instability during physiological aging. Rossi and colleagues tested these hypotheses in mice deficient in several genomic maintenance pathways including nucleotide excision repair, telomere maintenance, and non-homologous end-joining. DDR-deficient HSCs exhibited an accumulation of high levels of DNA double strand breaks (DSBs), reduced repopulation ability and self-renewal, as well as stem cell exhaustion, such as occurs in aged HSCs [1,40,42]. Similar to mouse HSCs, human CD34^+^ HSCs and hematopoietic progenitors showed a significant accumulation of DSBs during the normal aging process [43]. These reports strongly suggest that the appropriate DDR is important for the maintenance of HSCs and may prevent the functional decline of HSCs from aging.

### 3.2. ROS (Reactive Oxygen Species)

HSCs reside in hypoxic niches in the BM—an environment that presumably ensures HSCs are protected from oxidative stress and can maintain their self-renewal ability [31]. HSCs are quiescent, have a low metabolic rate, and therefore, generate low levels of reactive oxygen species (ROS). However, with aging, ROS levels accumulate and can result in ROS-induced oxidative stress in HSCs [1,44,45]. Aging increases ROS levels, which contribute to increased proliferation, senescence, or apoptosis. HSCs exposed to low ROS levels maintained a higher self-renewal potential in serial transplantation. However, HSCs exposed to high ROS levels failed to self-renew and showed higher levels of expression of activated p38 mitogen-activated protein kinase (p38) and mammalian target of rapamycin (mTOR) [46]. Porto and colleagues found that aged HSCs showed increased oxidative stress compared to young HSCs; mitochondria and NADPHox were the major sources of ROS production, whereas CYP450 contributed in middle and aged HSCs and xanthine oxidase only in aged HSCs [47]. Thus, oxidative stress might be considered an important cause of HSC dysfunction during the aging process [48]. We have reported the regulatory mechanisms of ROS in aged HSCs using specific gene knockout mice. We found that thioredoxin-interacting protein (TXNIP) regulated ROS levels in HSCs by regulating p53 activity via interference with p53-mouse double minute 2 (MDM2) interactions [5]. More recently, we revealed that TXNIP regulated p38 activity—a critical inducer of ROS in aged HSCs [19]. Recent reports have also demonstrated that ROS acted as critical regulators of HSC aging and go on to determine regulatory mechanisms. The Forkhead O (FOXO) subfamily of transcription factors—including Foxo1, Foxo3a, and Foxo4—regulated pools of HSCs and progenitors as well as ROS in HSCs [49,50]. Hypoxia-inducible factor-1α (HIF-1α) was highly expressed in HSCs. HIF-1α changed cellular metabolism of HSCs from mitochondrial respiration to glycolysis and thereby reduced ROS production. Deletion of HIF-1α in HSCs induced ROS levels and reduced long-term repopulation capacity [31,51]. Altogether, the importance of ROS as an inducer of aging in HSCs was determined by cellular regulatory mechanism studies.

### 3.3. Epigenetic Changes

Epigenetics refers to changes in gene expression that do not involve changes to the underlying DNA sequence, that is a change in phenotype without a change in genotype [52]. Analysis of gene transcriptional profiles indicated that myeloid differentiation-linked and inflammation- and stress response-related genes were upregulated, but lymphopoiesis, DNA repair, and chromatin silencing genes were downregulated in aged HSCs [13,53]. Aged HSCs exhibited broader H3K4me3 peaks across HSC identity and self-renewal genes, exhibiting increased DNA methylation at transcription factor binding sites associated with differentiation-promoting genes. There is a strong correlation between altered H3K4me3 levels and transcriptional activity, involving genes that undergo increased expression levels, most significantly with age [53]. These gene expression changes are in accordance with age-induced myeloid skewing, at the level of gene transcription. It also implies that HSC aging is related to epigenetic changes that impact transcriptional regulation. The function of DNA methylation includes the induction of physiological processes such as stem cell differentiation [52]. Genetic inactivation of DNA methyltransferase 1 (DNMT1), induced nearly immediate and complete loss of HSCs in vivo [54]. Reduced DNMT1 activity in HSCs restricted myeloerythroid differentiation due to both impaired silencing of key lineage determinant genes and an inability to prime master lymphoid regulators [55]. DNMT3a also regulates HSC fate decisions, as evidenced by the conditional inactivation of DNMT3a, which skews HSC divisions toward self-renewal at the expense of differentiation [56]. The expression of genes encoding DNA methyltransferase enzymes decreases in aged HSCs compared to young; moreover, aged HSCs are characterized by a gradual increase in global DNA methylation levels [53]. Corresponding with the DNA hypermethylation of aged HSCs, there is a concomitant reduction in 5-hmC in aged HSCs compared to young, with all three TET enzymes decreasing in abundance, with age [52,53]. Florian and colleagues demonstrated that young HSCs expressed higher levels of AcH4K16 and aged HSCs presented decreased levels of AcH4K16 [57]. However, these mechanisms are inadequate to explain the exact correlation between epigenetic changes and HSC aging; therefore, the functional consequences of epigenetic changes in HSCs remain to be determined.

### 3.4. Polarity Changes

The small RhoGTPase Cdc42 (Cdc42) cycles between an active (GTP-bound) and an inactive (GDP-bound) state and is known to regulate actin and tubulin organization, cell-cell and cell-extracellular matrix adhesion, and cell polarity in distinct cell types [57,58,59]. Florian and colleagues reported Cdc42 as a new aging marker for HSCs. They found that elevated Cdc42 activity in aged HSCs was causally linked to HSC aging and correlated with a loss of polarity in aged HSCs. Constitutively activated Cdc42 resulted in premature aging of HSCs and induced depolarization of Cdc42 and tubulin in aged HSCs. Pharmacological reduction of Cdc42 activity reversed the polarity of Cdc42 and tubulin and restored cellular function of aged HSCs [57]. We also demonstrated that aged HSCs showed depolarization of Cdc42 allowing us to use it as an aging marker for HSCs in our study [19].

## 4. Rejuvenation of Aged HSCs

### 4.1. Reduction of Nutrient Supply

Mutations in growth signaling pathways extend the life span while protecting against age-dependent DNA damage in yeast [60]. Prolonged fasting (PF) reduces progrowth signaling and activates pathways that enhance cellular resistance to toxins in mice and humans [60,61,62]. One group has suggested that caloric restriction by PF can rejuvenate aged HSCs. They reported that PF reduced circulating IGF-1 levels and PKA activity in various cell populations and rejuvenated the aging-associated phenotypes of aged HSCs including myeloid bias. PF also protected mice from chemotherapy-induced immunosuppression and mortality [63].

Reversible acetylation of metabolic enzymes regulates metabolic pathways in response to nutrient availability [64]. The sirtuin family are key regulators of the nutrient-sensitive metabolic regulatory circuit. Calorie restriction reduced ROS by inducing the activation of SIRT3-mediated superoxide dismutase 2 (SOD2) [65]. Brown and colleagues demonstrated that SIRT3 regulated the global acetylation of mitochondrial proteins, which are enriched in HSCs. The expression of SIRT3 decreased with aging; moreover, SIRT3 was dispensable for young HSCs but was essential under stress or for aged HSCs. Finally, the authors suggested that the plasticity of mitochondrial homeostasis controls HSC aging and showed that aged HSCs can be rejuvenated by SIRT3 expression [66]. Mohrin and colleagues have reported on the interaction between SIRT7 and nuclear respiratory factor 1 (NRF1). SIRT7-deficient HSCs showed a reduction in repopulating capacity and myeloid-biased differentiation in transplantation assays. The upregulation of SIRT7 improved the regenerative capacity of aged HSCs [67].

### 4.2. ROS Scavenging

HSCs exhibit low metabolic rates and produce less ROS. The ataxia telangiectasia mutated (ATM) gene is essential for HSC self-renewal and quiescence; ATM regulates ROS levels in HSCs. ATM-deficient mice showed a defect in HSC function and elevated ROS levels in HSCs. Treatment with the ROS scavenging agent, N-acetyl-L-cysteine (NAC), rescued the repopulating capacity of ATM-deficient HSCs [68]. p38 limits the lifespan of HSCs in vivo by inducing ROS; the inhibition of p38 by treatment with SB203580, a p38 inhibitor, rescued ROS-induced defects in HSC repopulating capacity and HSC quiescence maintenance [69]. Recently, we developed a new p38 inhibitor—TN13—which is a cell-penetrating peptide-(CPP-), a conjugated peptide derived from the TXNIP-p38 interaction motif in TXNIP. TN13 inhibited p38 activity and rejuvenated aged HSCs by reducing ROS levels [19]. G protein-coupled receptor kinases (GRKs) regulate cytokine receptors in mature leukocytes. GRK6^−/−^ mice exhibited HSC loss and impaired HSC self-renewal. GRK6 also regulates ROS response, and ROS scavenger α-lipoic acid treatment rescued HSC loss in GRK6^−/−^ mice [70]. 

### 4.3. Epigenetic Modulation

Elevated Cdc42 activity is associated with HSC aging and induces decreased levels of H4K16Ac in aged HSCs. Treatment with a specific inhibitor of Cdc42 activity (CASIN) can restore aged HSC phenotypes by regulating both Cdc42 activity and epigenetic reprograming by elevating H4K16Ac levels to those of young cells [1,57]. Special AT-rich sequence binding 1 (Satb1), a nuclear architectural protein, regulates chromatin structure by epigenetic regulation and plays an important role in lymphoid lineage specification. The expression of Satb1 increased with early lymphoid differentiation and Satb1-deficient HSCs showed a reduction in lymphopoiesis, while induced Satb1 expression enhanced lymphocyte production. Exogenous Satb1 expression primed the lymphoid potential of aged HSCs [71].

### 4.4. Clearance of Senescent Cells

Aging and genotoxic stress induce cellular senescence of HSCs [72,73]. Senescent cells (SCs) accumulate in various tissues and organs with aging, thereby disrupting tissue structure and function by secreting cellular components. Clearance of SCs in a mouse model using a transgenic approach revealed delays in several age-associated disorders [74]. Chang and colleagues developed a potent senolytic drug ABT263—a specific inhibitor of BCL-2 and BCL-xL, that selectively kills SCs. Oral administration of ABT263 effectively depleted SCs—including senescent HSCs in both sub-lethally irradiated and normally aged mice. Depletion of SCs mitigated irradiation-induced premature aging of the hematopoietic system and rejuvenated aged HSCs in normally aged mice [75]. Given these results, Chang et al. determined that ABT263 may represent a new class of radiation mitigators and anti-aging agents.

## 5. Emerging Technologies for HSC Study

### 5.1. Single-Cell RNA-Sequencing (scRNA-Seq)

HSCs are heterogeneous and HSC fate decisions are performed at the individual cell level. Thus, single-cell profiling may be an essential tool for characterizing the heterogeneity of HSCs. Most recently, single-cell RNA-sequencing (scRNA-Seq) was developed to profile single cells and was applied in the studies of HSCs [76]. Kowalczyk and colleagues used scRNA-Seq to dissect variability in HSCs and HPCs from young and old mice. They found that transcriptional changes in HSCs during aging are inversely related to those upon HSC differentiation, and old ST-HSCs exist in a less differentiated state than young ST-HSCs [77]. One group have revealed that the megakaryocyte lineage is developed independently of other hematopoietic fates using scRNA-Seq. They have also found a functional hierarchy of unilineage- and oligolineage-producing clones within the MPPs [78]. Grover and colleagues identified an unrecognized class of HSCs that exclusively produce platelets with age. They suggested that increased platelet bias may contribute to the age-associated decrease in lymphopoiesis [79]. More recently, Baron and colleagues reported the genes and transcription factor networks activated during the endothelial-to-haematopoietic switch using scRNA-Seq. From this study, they provide an unprecedented complete resource to study in depth HSC generation in vivo [80].

### 5.2. Single-Cell Transplantation

HSC transplantation assays evaluate the capacity of HSCs to repopulate the hematopoietic system. HSC heterogeneity was initially identified by tracking of myeloid and lymphoid lineage output from transplanted single HSCs [81,82,83]. Single-cell transplantation has shown considerable heterogeneity of HSCs and different propensities of myeloid-, lymphoid- and platelet-biased HSCs in fate decision [84]. Carrelha and colleagues applied single-cell transplantation assays to investigate lineage-restricted fates of long-term self-renewing HSCs. They found that a distinct class of HSCs adopts a fate towards megakaryocyte/platelet-lineage tree, but not of other blood cell lineages. Finally, they suggested that a limited repertoire of distinct HSC subsets adopt a fate towards replenishment of a restricted set of blood lineages, before loss of self-renewal and multipotency [84]. Yamamoto and colleagues investigated the age-related functional changes in HSCs using single-cell transplantation assays. They found that myeloid-restricted repopulating progenitor (MyRPs) frequency increased dramatically with age. Interestingly, they identified new subsets, latent-HSCs, that were myeloid restricted in primary recipients but displayed multipotent (five blood-lineage) output in secondary recipients. From these results, they have raised a question about the traditional dogma of HSC aging and our current approaches to assay [85].

## 6. Conclusions

Here, we introduced the hallmarks of HSC aging including defects in repopulation capacity, homing, and an increase in mobilization and myeloid-biased skewing. HSC aging is driven by multiple cell-intrinsic factors such as DNA damage, ROS, epigenetic changes and polarity changes, and cell-extrinsic factors including cytokines and chemokines derived from the HSC niche. We also mentioned recent emerging technologies for single-HSC profiling such as scRNA-Seq and single-cell transplantation. These methods have proven highly valuable in the unravelling of HSC heterogeneity, as described above. Surprisingly, aged HSCs could be rejuvenated by reduction of the nutrient supply, ROS scavenging, epigenetic modulation or the clearance of senescent cells (Figure 2). Although we have described the defined mechanisms and the physiological or biological changes involved in HSC aging, this does not explain all the aspects of aging, many of which remain elusive. However, our accumulated knowledge may be helpful for understanding HSC aging and designing HSC aging studies. Fortunately, many HSC aging studies have suggested that the dysregulated functions of aged HSCs can be reversed or rescued by rejuvenating agents or the overexpression of specific genes. In the future, these approaches may shine a light on treating age-related immune diseases and can also provide a promising tool to improve quality of life for the elderly.

## Figures and Tables

**Figure 1 ijms-20-01272-f001:**
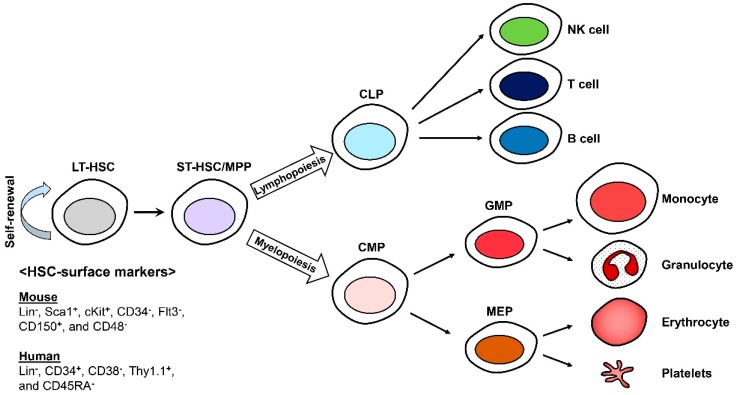
Differentiation of hematopoietic stem cells (HSCs). Long-term HSCs (LT-HSCs) are able to self-renew and are responsible for generating blood cells. CLP; the common lymphoid progenitor, CMP; the common myeloid progenitor, GMP; the granulocyte macrophage progenitor, and MEP; the megakaryocytic and erythroid progenitor.

**Figure 2 ijms-20-01272-f002:**
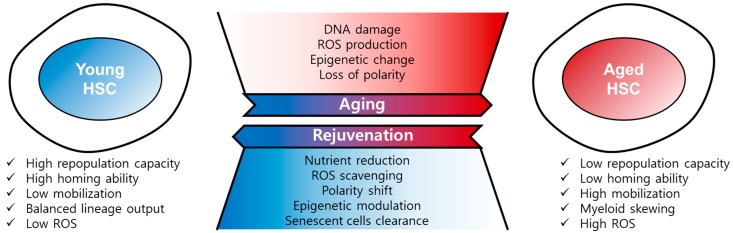
Regulation of HSC aging and rejuvenation. Aged HSCs have the hallmarks of low repopulation capacity, low homing ability, high mobility, myeloid skewing, and high ROS, among others. HSC aging is driven by DNA damage, ROS, epigenetic change, and loss of polarity. Aged HSCs can be rejuvenated by nutrient reduction, ROS scavenging, polarity shift, epigenetic modulation, and senescent cells clearance.

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
