# Peer review of "Causes and Mechanisms of Hematopoietic Stem Cell Aging"

_ijms, 2019, doi:10.3390/ijms20061272_

Round 1
Reviewer 1 Report
Causes and mechanisms of stem cell aging
The review by Lee et. al., presents phenotype or characteristics of aging HSCs, causes and possible modes in which this can be changed. The review is clear and concise and easy to read. There are some suggestions which the authors might consider:
1. The review is divided into 4 sections. The authors should introduce what they are presenting in these sections in the introduction so that the reader knows what to expect.
2. The authors mention cell-intrinsic and extrinsic factors affecting aging in both the abstract and introduction. The review does not specifically substantiate which causes described in the review are cell extrinsic/intrinsic. Provide references to support your statement in the introduction and it would be appropriate based on the discussion in the review to mention which factors are cell-intrinsic and extrinsic in “Conclusions”.
3. Section 3.5 Myeloid biased clonal expansion seems out of place in the category of “Causes”. This is more of a phenotype which has already been explained in section 2.3 “Lineage Skewing”. Section 3.5 is not adding anything specific to the review and can be removed.
4. Lines 82-83 seemed confusing. The authors state that in G-CSF-receptor (G-CSFR)-deficient mice, all hematopoietic cells completely failed to mobilize. Why is that? Provide some references to support this because the lines from 83 onwards do not explain the reason for failed mobilization.
5. Wherever possible please reconstruct longer sentences into short ones for easy understanding. For example line 142-145 is too long and needs re-reading to make sense.
6. There are several papers with single cell RNA sequencing and single cell transplantation to understand the issue of stem cell heterogeneity. The review does not make any mention of this. If there is enough evidence please present this as a small section. If the evidence is not substantial then at least make a mention in a few lines stating the most prominent work.
7. Please provide at least one more illustration to make the review more presentable and interesting.
Author Response
Reviewer-1
Comments and Suggestions for Authors
Causes and mechanisms of stem cell aging
The review by Lee et. al., presents phenotype or characteristics of aging HSCs, causes and possible modes in which this can be changed. The review is clear and concise and easy to read. There are some suggestions which the authors might consider:
Response:
We appreciate the reviewer’s comments. We have added one more figure for the differentiation of HSCs. We have rearranged the manuscript as reviewers mentioned. Please check a new manuscript version and modified manuscript was clearly highlighted by “track changes” function in Microsoft Word.
The review is divided into 4 sections. The authors should introduce what they are presenting in these sections in the introduction so that the reader knows what to expect.
Response:
As reviewer mentioned, we have added a sentence for section titles in the introduction.
The authors mention cell-intrinsic and extrinsic factors affecting aging in both the abstract and introduction. The review does not specifically substantiate which causes described in the review are cell extrinsic/intrinsic. Provide references to support your statement in the introduction and it would be appropriate based on the discussion in the review to mention which factors are cell-intrinsic and extrinsic in “Conclusions”.
Response:
We have described about cell-intrinsic and extrinsic factors affecting aging in the introduction and Conclusions, and added references.
Section 3.5 Myeloid biased clonal expansion seems out of place in the category of “Causes”. This is more of a phenotype which has already been explained in section 2.3 “Lineage Skewing”. Section 3.5 is not adding anything specific to the review and can be removed.
Response:
As reviewer suggested, we have removed section 3.5.
Lines 82-83 seemed confusing. The authors state that in G-CSF-receptor (G-CSFR)-deficient mice, all hematopoietic cells completely failed to mobilize. Why is that? Provide some references to support this because the lines from 83 onwards do not explain the reason for failed mobilization.
Response:
We have edited the sentences and supporting references are reference number 30 and 31.
Wherever possible please reconstruct longer sentences into short ones for easy understanding. For example line 142-145 is too long and needs re-reading to make sense.
Response:
Including line 142-145, we have reconstructed longer sentences into short ones in the manuscript.
There are several papers with single cell RNA sequencing and single cell transplantation to understand the issue of stem cell heterogeneity. The review does not make any mention of this. If there is enough evidence please present this as a small section. If the evidence is not substantial then at least make a mention in a few lines stating the most prominent work.
Response:
I have reviewed recent reports for single cell RNA sequencing and single cell transplantation and then added a new section (Section 5. Emerging technologies for HSC study) for them.
Please provide at least one more illustration to make the review more presentable and interesting.
Response:
We have prepared one more illustration for the differentiation of HSCs (Figure 1).

Reviewer 2 Report
The article is well written and a nice summary of the field. Well done.
I went through the paper fairly carefully. The grammar is very good and the references are accurate. Aside from a few long sentences, which the other reviewer has pointed out, the article is easy to understand. For the length of the manuscript, I think they did a good job of summarising the topic of ageing in HSC. The authors have added some of their own personal research and perspective, which I think is a fair reason to write a review and showcase how your work fits into the broader field. I think the sections are well organised. The authors could include additional figures to summarise some of the other sections and improve the comprehension of the article visually, if there's room for more more figures.
Author Response
Reviewer-2
Comments and Suggestions for Authors
The article is well written and a nice summary of the field. Well done.
I went through the paper fairly carefully. The grammar is very good and the references are accurate. Aside from a few long sentences, which the other reviewer has pointed out, the article is easy to understand. For the length of the manuscript, I think they did a good job of summarising the topic of ageing in HSC. The authors have added some of their own personal research and perspective, which I think is a fair reason to write a review and showcase how your work fits into the broader field. I think the sections are well organised. The authors could include additional figures to summarise some of the other sections and improve the comprehension of the article visually, if there's room for more figures.
Response:
We appreciate the reviewer’s comments. We have added one more figure for the differentiation of HSCs and Reviewer 1 has also raised some comments for the manuscript. We have rearranged the manuscript as reviewers mentioned. Please check a new manuscript version and modified manuscript was clearly highlighted by “track changes” function in Microsoft Word.
